# Quality of Work Life of Magnetic Resonance Imaging Technologists: A Cross-Sectional Study

**DOI:** 10.3390/healthcare10122539

**Published:** 2022-12-15

**Authors:** Ghadah F. Almugren, Haya S. Zedan

**Affiliations:** 1Department of Medical Imaging, Ministry of National Guard Health Affairs, King Abdullah International Medical Research Center, King Saud bin Abdulaziz University for Health Sciences, Riyadh 11426, Saudi Arabia; 2Department of Public Health, College of Health Sciences, Saudi Electronic University, Riyadh 13316, Saudi Arabia

**Keywords:** quality of work life, work–life balance, healthcare, magnetic resonance imaging, human resources, health care quality

## Abstract

Quality of Work Life is a multi-dimensional discipline that is concerned with the quality of life in the workplace. This study aimed to assess quality of work–life level and identify the correlation between its dimensions and Job and Career Satisfaction. The study used the 32-item WRQoL-2 tool, a questionnaire consisting of 6 subscales: Job and Career Satisfaction, Control at Work, Home–Work Interface, General Wellbeing, Stress at Work, and Work Conditions, to assess for these correlations. 57 Magnetic Resonance Imaging Technologists (MRITs) (100%) responded to the questionnaire. The study found a high level of QWL among MRITs (66.2%, 3.31/5). The level of the JCS was high (71.6%, 3.59/5), with significant correlations between JCS and WCS, CAW, HWI, and GWB. An inverse relationship was noted between SAW and JCS. Further research on QWL is advised to diagnose and provide recommendation to resolve issues that may adversely affect the quality of healthcare service provision.

## 1. Background

Quality of work–life (QWL) is defined as the degree of satisfaction that employees achieve regarding working conditions, remuneration, professional development, work-family role balance, safety and social interactions in the workplace, and social relativity of work [1]. QWL is the staff’s satisfaction derived from work life. It is a subjective phenomenon influenced by personal feelings and perceptions [1,2,3]. QWL is an interdisciplinary area incorporating elements of psychology, sociology, and healthcare management. It combines strategies and procedures within a workplace, aiming to support staff status and enhance work–life conditions. QWL has evolved since the 1960s, intending to create homogeneity between staff and the work environment. Louis Davis first introduced the term “Quality of Work Life” (QWL) in 1972 [1]. Recently, Easton and Van Laar [3] developed the Work-Related Quality of Life (WRQoL-2) Scale to assess QWL, which is comprised of 6 subscales: Job and Career Satisfaction (JCS), Control at Work (CAW), Homework Interface (HWI), General Wellbeing (GWB), Stress at Work (SAW) and Work Conditions (WCS).

QWL is a broad concept that considers staff wellbeing, work experiences, and the work environment to achieve success and the organization’s objectives. Behavioral scientists agree on eight essential categories of QWL: (1) safe and secure working conditions, (2) adequate pay and compensation, (3) skills development, (4) continuous growth, (5) social integration within the organization, (6) work and total living space, (7) constitutionalism in the organization and (8) social relevance to work–life [1]. QWL has the potential to increase staff professional growth, productivity, and achievement [2,4,5]. The concept has numerous dimensions: work–life balance, work environment, participative management, job satisfaction, reward and recognition, proper grievance handling, welfare facilities, and organizational commitment [5]. 

Differentiating career satisfaction and job satisfaction is crucial to understand QWL. Job satisfaction is a subjective condition that involves happiness arising from job responsibility, independence, and work experience. Job satisfaction relates to the achievement and maintenance of success. However, job dissatisfaction is the major cause of large team turnover and defection [2,6,7]. There is an inverse relationship between job satisfaction, absenteeism, turnover rate, conscientiousness, and staff commitment [2,6,7]. When job satisfaction is high, absenteeism and turnover rate is low. When job satisfaction is low, absenteeism and turnover rate is high. A model of job satisfaction comprises of four elements: job performance, company performance, role perceptions, and job-related factors [6,7,8]. Four variables that may affect the perception of staff QWL are meaningfulness, pessimism regarding a change in the organization, job satisfaction, and self-confidence [9]. These four qualities of work–life variables were positively correlated with staff perception of job satisfaction. 

Career satisfaction is defined as the staff’s wellbeing in the physical and external dimensions of the work, incorporating career progression and opportunities for the staff [10]. Staff report a high level of QWL when they experience low conflict and pressure, are satisfied with their work, and enjoy social connections within the work environment.

The QWL conceptual framework is viewed from four dimensions: work environment, work context, work life/home, and life–work design [5]. The work environment is defined as the impact of societal changes on professional practice. The work context incorporates the work setting and the effect of the work environment on staff. The work–life/home life dimension is the integration and compromise between home life and work life. The life–work design deals with the composition of the work and identifies the actual work responsibilities. For instance, the availability of work tasks, security, education, staffing ratios, and individual success are part of the quality of the work–life equation [11]. 

Excellent work conditions and low work stress are critical to improving productivity and job satisfaction. The best condition reduces turnover and enhances robust QWL, which provides a conducive environment to attract new staff and increase staff commitment. Thus, QWL enhances team performance, reduces absenteeism, reduces burnout, and lowers staff turnover rates. The concept also enhances work conditions and encourages staff to be more active and vital within the organization. Therefore, policies and procedures that create a positive work experience [7,9,12] support staff satisfaction with tasks. Policies and procedures could consist of elements to promote independence, loyalty, skill development, external rewards, and giving credit for the work. Staff’s QWL can positively influence human resources recruitment and staffing. Thus, organizations can use QWL to satisfy staff demands and meet their needs both inside and outside the work environment [10,13]. Organizations can also use the QWL model to bolster staff strength, reduce stress, meet performance requirements, and improve experiences at work [13]. Supportive visionary leadership is crucial to set positive QWL by developing effective management resources and raising collaboration between staff [13,14].

As the healthcare work environment affects healthcare service quality, healthcare administrators must provide the best quality work environment to achieve the organizations’ vision, mission, and goals [14]. Globally, there is an increasing interest in QWL levels bolstering organizational performance and improving quality of patient care [12]. A high level of QWL is essential for healthcare organizations, to attract and retain talented and qualified healthcare personnel. High levels of QWL facilitate staff commitment, engagement, and satisfaction. For example, healthcare organizations that manage highly technical professionals must take QWL into high consideration because high performance is the core of organizational success, in turn impacting patients’ satisfaction and wellbeing [15]. As retention is critical to organizational performance, many healthcare organizations are investigating issues of retention and recruitment to accomplish a high QWL aesthetic [2,12,16,17].

In another dimension, the quality of attention given to patients significantly correlates to the QWL of healthcare personnel [14,15]. Job satisfaction has been linked to raising the quality of service in the healthcare setting. It is also linked to leadership and work policies, each of which has a contemplative impact on how staff view their workplace. QWL aims to promote and sustain staff satisfaction, thereby increasing productivity and accomplishing the organization’s objectives [7,8,14].

A growing body of research has evaluated the impact of QWL on healthcare professionals [2,4,7,11,12,14,15,16]. However, no studies have dealt with QWL levels for Magnetic Resonance Imaging Technologists (MRITs) in the Kingdom of Saudi Arabia (KSA). A thorough search of the literature in Medline, CINAHL, Web of Science, and PubMed yielded no results. Thus, this study reviewed published literature that focused on QWL and appreciating the similarities of the nurse’s work environment and MRITs. This study focused on work conditions, pressure, and working hours. Powerful QWL can influence staff commitment and contribution within the healthcare setting.

Several studies have explored QWL in the medical sector in KSA, aiming to identify the relationship between QWL and turnover intention of nurses [16,17,18]. Brooks’ scale for Quality of Nursing Work Life (QNWL) was used as a tool to collect data. 508 nurses in the Jizan region of the Kingdom of Saudi Arabia were surveyed, and they found dissatisfaction among the respondents regarding their QWL. A total of 203 nurses expressed an intent to leave their jobs—a potential turnover rate of 40%. The average score on the Brooks’ scale for respondents was (139.45/252), a low QWL score [16]. The same scale (QNWL) was used to evaluate QWL and nursing turnover intention for 364 nurses working in King Fahad Medical City and King Faisal Specialist Hospital and Research Center [17]. From the results, 54.7% of nursing staff expressed dissatisfaction with QWL in both hospitals. The turnover intention rate of 94% was very high for both hospitals. A study aimed to measure the correlation between QWL and work engagement in the Kingdom [18]. The sample consisted of 207 nurses working in hospitals in the Eastern region. The results showed a correlation between the three dimensions of job engagement: absorption, dedication, and vigor [18].

MRITs are qualified healthcare staff tasked to manage radiology imaging equipment, which doctors use to diagnose and treat patients. This current study used a similar methodology previously used to study QWL in nursing staff. MRITs and nurses have similar working hours, experience similar work conditions, and face similar challenges and pressures in their work environments. The two types of professionals are responsible to complete an intense workload. They also interact with multiple specialists and face delays in work practice, as well as performing non-work-related duties such as supporting family. Other challenges they face in career progression are training, financial benefits, flexible scheduling, and other issues potentially impacting their QWL.

### Purpose of the Study

This study assessed the correlation between QWL and JCS for MRITs working at National Guard Health Affairs Hospitals (NGHA). The goal was to measure the QWL and explore the potential to raise the quality of healthcare services and patients’ satisfaction as a result.

## 2. Methods

### 2.1. Study Setting

NGHA is a division of the Ministry of the National Guard, one of the largest medical organizations in the KSA. NGHA is distinguished by its high-quality imaging departments in six hospitals across the cities of Riyadh, Dammam, Jeddah, Al-Ahsa, and Al-Madinah Al-Munawarah.

### 2.2. Design/Methodology/Approach

This study used a cross-sectional design between September and December 2019, using the WRQoL-2 to collect data [3]. WRQoL is a scale initially developed to collect data about QWL in medical environments [19]. The scale consisted of 24 items in 6 subscales. It was increased to 32 items in 2018 to provide better psychometric properties [3]. The 6 subscales of WRQoL-2 are Job and Career Satisfaction (JCS), Control at Work (CAW), Home–Work Interface (HWI), General Wellbeing (GWB), Stress at Work (SAW), and Work Conditions (WCS). The 32 items are distributed to the six subscales as follows: items no. (1.3.8.11.18.20) to JCS, items no. (2,12,23,30) to CAW, items no. (5,6,14,25) to HWI, items no. (4,9,10,15,17,21,27,28) to GWB, items no. (7,19,24,29) to SAW, and items (13, 16, 22, 26, 31) to WCS. Respondents used a five-point Likert scale (5 = strongly disagree, 1 = strongly agree). An electronic questionnaire was developed from the WRQoL-2. With permission, it was sent to study respondents through the (WhatsApp) mobile application.

### 2.3. Study Sample

The study sample consisted of all the MRITs, male or female, Saudi or non-Saudi, and staff on the job training under the Saudi Career Development Program in all six hospitals located within the NGHA network. Upon enquiry from the NGHA Human Resources Department, a full list of all MRITs at the organization (n = 57) was established. Magnetic resonance imaging (MRI) is a highly specialized field, therefore the number of respondents who are specialized and qualified in this field is understandably small. This also makes this current study more relevant, as the QWL of this specialization may be robustly linked to the quality-of-service provision and satisfaction for patients.

### 2.4. Data Analysis

Statistical tests were conducted to establish the sample distribution; it was found to be normal. Frequencies and percentages were calculated to describe study sample demographics. The mean was calculated to find high and low responses of the study respondents on each subscale. A One-Sample *t*-test was used to determine whether the average score of the items is higher or lower than the average approval score, which is 3. The standard deviation was used to identify the extent of deviation from responses in each subscale.

Validity and reliability tests were also conducted. The Pearson correlation coefficient was used to measure the validity of the study tool. The Alpha Cronbach coefficient was used to measure the reliability of the study tool. The entire scale items and their correlations with subscales were significant at (*p* < 0.01), denoting that all questionnaire items have internal consistency and inter-items consistency. To check the reliability of the study tool, the Cronbach α coefficient of 0.9 was calculated for the WRQoL-2 six subscales. The study tool was therefore highly reliable to measure the QWL for MRITs.

## 3. Results

N = 57 of MRITs surveyed responded (100%). Of the respondents, 53.6%, n = 30 were male and 47.4%, n = 27 were female. 84% of respondents were in the age group 25–44 years (n = 48), followed by the age group 45–59 years, n = 6 (11%). Only three respondents (5%) were under 25 years of age.

Of the respondents, n = 26 (45.6%) have 1–5 years of experience, representing the largest group, followed by respondents who have 6–10 years of experience, n = 14 (24%). Those with 11–20 years of experience are n = 11 (19%). Only 2 (3.5%) respondents had more than 20 years of experience. A total of n = 40 (70.2%) respondents work on a full-time basis with paid overtime, which denotes a shortage of the MRITs in NGHA hospitals. Seventeen (29.8%) respondents work full-time with no overtime. A total of 56 (98%) respondents did not have any disabilities; one (1.8%) respondent had a disability.

A total of 38 (67%) respondents work in NGHA Riyadh, representing the highest proportion, followed by n = 6 (11%) in Jeddah and Madinah each. N = 4 (7%) respondents work in Dammam, and n = 3 (4%) respondents work in Al-Ahsa. A total of 21 (36.8%) respondents did not have dependents. N = 13 (22.8%) respondents have responsibilities with school-age children, and n = 8 (14%) respondents have babies and young children. Four respondents (7%) have elderly-dependent relatives (Table 1).

Mean and standard deviation were calculated for the study responses. The One-Sample *t*-test was used to identify the degree of QWL for each subscale and the overall degree of QWL for MRITs. The study sample of MRITs was found to have a high level of QWL. (Table 2). The mean for the study responses was (3.31/5.00). Four subscales were statistically significant at the level of (*p* < 0.01). The study responses were higher than the average approval level of (3), which confirms that QWL for MRITs was high. The highest scores among the six WRQoL-2 dimensions were for WCS, followed by GWB, SAW, and HWI. All four dimensions were high for MRITs, the mean ranging between 3.28 and 3.60. The evaluation rate was between 65.6% and 70.2%. All were statistically significant at (*p* < 0.05).

The One-Sample T-test was conducted for statements 1, 3, 8, 11, 18, 20, corresponding to JCS on the WRQoL-2 Scale. The mean of the responses was (3.59/5), indicating a high level of JCS for the study sample. This was significantly correlated at (*p* < 0.01) (Table 3).

Using Pearson’s Correlation Coefficient, there was a positive correlation between the JCS and the other subscales of WRQoL-2 (GWB, HWI, CAW, WCS, and SAW) (Table 4). A statistically significant correlation is established between the JCS and all subscales at the (*p* < 0.01) level, revealing that when work conditions, wellbeing, and control at work improve, an increase in job and career satisfaction is recorded. In addition, an inverse correlation between SAW and JCS is recorded as the significance level was at a negative (*p* < 0.01) level. When “Stress at Work” increases, JCS decreases.

No statistically significant differences at the level of significance (*p* < 0.05) are recorded for the subscales HWI, CAW, WCS, and SAW. The overall WRQoL-2 score was not statically significant for gender. We found statistically significant differences for the subscales JCS, GWB according to the gender variable (*p* < 0.01) in favor of the male gender (Table 5).

Using the ANOVA test, there was a statistically significant difference between the overall level of WRQoL-2 and age. The HWI, CAW, and WCS subscales were statistically significant by age group at (*p* < 0.01). The JCS, GWB, and SAW subscales did not offer any significance (Table 6).

Scheffe’s post hoc test was conducted to identify differences between age groups for the HWI, CAW, and the WCS subscales (Table 7).

A significant difference is noted among the study respondents concerning HWI, CAW, WCS in the 45–59 years group, indicating a correlation between QWL and older age.

## 4. Discussion

The level of QWL for MRITs at NGHA hospitals was assessed and identified the JCS and differences between gender and age in the six WRQoL-2 subscales. Many studies, conducted in China, Taiwan, Turkey, Iran, Uganda, and the UK, [20,21,22,23,24,25,26,27] have used WRQoL-2. These studies identified problems and factors that affected QWL to enhance the working conditions and the working environment. This is crucial as high levels of QWL contribute to productivity, excellence and quality of service provision at a workplace.

In comparison to the findings of other studies, this current study shows that job and career satisfaction in MRITs at NGHA hospitals is highest (3.59), followed by the study on nurses in Taiwan (3.75) [22], and followed by the study on nurses in Uganda (3.53) [23]. Concerning CAW, three studies scored (3.39), and the current study came second with a score of (3.15/5). This indicates a similarity between all the studies that used the WRQoL-2 tool for the CAW dimension. In the current study, the HWI dimension score was (3.28/5), whereas the score by [24,25] was (3.52) and (2/5) by [26]. For the GWB dimension, the current study scores (3.51), which is the highest among the nine studies (Table 8).

Thus, the wellbeing of medical staff in KSA is considerably higher than healthcare personnel in other developing countries. The overall results of the current study show that the MRITs working in NGHA hospitals have a high degree of JCS; the study respondents are satisfied with their jobs.

In this study, the condition of the work of the nursing staff differed slightly from that of the previous study focusing on Al-Madinah region hospitals in KSA [27]. This was because the study on Al-Madinah region hospitals focused on a single region, whereas the current study was conducted across KSA. The mean for the work–life/home dimension in the study focusing on the Al-Madinah region was (3.37). However, in the current study, the mean HWI is (3.28).

In this study, female respondents scored low for General Wellbeing (3.31/5), compared with the results of other studies that scored low for General Wellbeing of females (3.23/5) [28]. The overall WRQoL scores were similar for females in both studies (3.19). For the present study, the mean is 3.22. The results confirm our assumption that the MRITs face similar work environment issues to that of other healthcare professionals, and they face similar issues regarding their QWL. Male gender respondents were found to have higher mean scores than females in terms of job satisfaction. Thus, it showed that the total WRQoL contradicted a previous study that found moderate QWL among nurses [29]. The SAW dimension was found to be unrelated to gender, as male and female respondents had proportionate stress at work.

In addition, this study found that age and gender were significantly correlated with QWL. Respondents in the age group (45–59) had higher levels of WRQoL in the HWI dimension. The result is consistent with the findings of previous studies that older age has a positive effect on QNWL in the work environment [17,27]. Thus, older and more senior staff with years of work experience can adapt to their home and work lives, creating a balance and feeling satisfied with their roles. Their ability to overcome hardships, achieve job promotions, and ask for better pay is high.

The results of this study differ slightly from those in a study published in 2012, which found QWL dissatisfaction [16]. The 2012 study was conducted in a single region in the south of the Kingdom, where there was a shortage of a medical workforce, which led to increased workplace stress. Many studies measured QWL among other professions in the KSA. A study conducted on Yanbu industrial city staff found respondents had a high level of job satisfaction. The study identified factors that affected the level of job satisfaction, such as wages and remuneration, workgroup factors, and decision-making factors [30].

### Limitations and Recommendations

As discussed in the methods section, the study was limited to the small number of MRITs working in the NGHA system as a highly specialized group. While this is highly relevant to the field, there is definite potential for the WRQoL-2 to be applied to other groups of healthcare specialists or medical technologists in exploring QoL and its relationship to productivity, excellence and quality of healthcare service provision.

## 5. Conclusions

The current study aimed to assess the level of QWL, degree of job, and career satisfaction for MRITs working in NGHA. The study showed a high level of QWL and significant correlations between QWL and specific subscales. Older age and years of work experience contributed significantly to higher QWL scores. Exploring aspects of QWL for healthcare professionals and identifying shortcomings will help healthcare administrators bridge the gaps in this field, to further build and sustain the capacity of healthcare professionals.

## Figures and Tables

**Table 1 healthcare-10-02539-t001:** Sociodemographic Characteristics of Respondents.

	Frequency	Percent
Gender		
Male	30	43.6
Female	27	47.4
Age		
Under 25	3	5.3
25 to 44	48	84.2
45 to 59	6	10.5
Experience		
Less than 1	4	7.0
1 to 5	26	45.6
6 to 10	14	24.6
11 to 20	11	19.3
More than 20	2	3.5
Hours of Work		
Full time	40	70.2
Full time and paid overtime	17	29.8
Having a Disability		
Yes	1	1.8
No	56	98.2
Location of Work		
Riyadh	38	66.7
Jeddah	6	10.5
Al Ahsa	3	5.3
Dammam	4	7.0
Al Madinah	6	10.5
Dependency		
Babies/young children (under school age)	8	14.0
Babies/young children (under school age), School-age children	2	3.5
Babies/young children (under school age), School-age children, Disabled relatives, Elderly relative	4	7.0
Elderly relatives/friends	1	1.8
No	21	36.8
Other	4	7.0
School-age children	13	22.8
School-age children, Elderly relatives/friends	2	3.5
School-age children, Other	2	3.5
Babies/young children (under school age)	8	14.0
Total	57	100.0

**Table 2 healthcare-10-02539-t002:** QWL By Subscale.

One-Sample Statistic Test Value = 3				Sig. (2-Tailed)	Rate	Evaluation	Ordinal
Subscale	Mean	Std. Dev.	*t*				
WCS	3.60	0.71475	6.301	0.000 **	71.9	High	1
GWB	3.51	0.60328	6.404	0.000 **	70.2	High	2
SAW	3.42	0.92480	3.437	0.001 **	68.4	High	3
HWI	3.28	0.92965	2.280	0.026 *	65.6	High	4
CAW	3.15	0.79437	1.390	0.170 (Not sig.)	62.9	Moderate	5
Total Score n = 57	3.31	0.58907	3.994	0.000	66.2	High	

Test Value = 3; ** significant at 0.01; * sig. at 0.05.

**Table 3 healthcare-10-02539-t003:** T-test for JCS Statements.

One-Sample Statistics					(%)	Rating	Ordinal
Statement	Mean	Std. Dev	*t*	*p*-Value			
1. I have a clear set of goals and aims to enable me to do my job	4.12	0.908	9.339	0.00 **	82.4	High	1
3. I have the opportunity to use my abilities at work	4	0.707	10.677	0.00 **	80	High	2
11. I am encouraged to develop new skills	3.51	1.104	3.48	0.001 **	70.2	High	3
20. I am satisfied with the training I receive in order to perform my present job	3.42	1.085	2.931	0.005 **	68.4	High	4
18. I am satisfied with the career opportunities available for me here	3.4	1.1	2.77	0.008 **	68	High	5
8. When I have done a good job, it is acknowledged by my line manager	3.04	1.017	0.26	0.795	60.8	Moderate	6
Overall value of Job and Career Satisfaction	3.59	0.687	6.399	0.00 **	71.6	High	

** significant at 0.01.

**Table 4 healthcare-10-02539-t004:** Correlation Between QWL and JCS.

Subscales		GWB	HWI	CAW	WCS	SAW	Overall Score of QWL
JSC	Pearson Correlation	0.824 **	0.646 **	0.545 **	0.614 **	−0.339 **	0.802 **
	Sig. (2-tailed)	0.000	0.000	0.000	0.000	0.010	0.000

** significant at 0.01 level (2-tailed).

**Table 5 healthcare-10-02539-t005:** WRQoL-2 Subscales According to Gender Variable.

Subscales	Gender	N	Mean	Std. Dev	t	df	Sig. (2-Tailed)
JCS	Male	30	3.85	0.684	3.386	55	0.001 **
	Female	27	3.28	0.564			
GWB	Male	30	3.69	0.643	2.438	55	0.018 **
	Female	27	3.31	0.496			
HWI	Male	30	3.43	0.999	1.315	55	0.194
	Female	27	3.11	0.832			
CAW	Male	30	3.14	0.900	−0.017	55	0.986
	Female	27	3.15	0.675			
WCS	Male	30	3.68	0.814	0.904	55	0.370
	Female	27	3.51	0.587			
SAW	Male	30	3.38	0.980	−0.322	55	0.749
	Female	27	3.46	0.876			
Overall WRQoL Score	Male	30	3.42	0.641	1.474	55	0.146
	Female	27	3.19	0.511			

** Significant at 0.01.

**Table 6 healthcare-10-02539-t006:** WRQoL-2 Subscales According to Age Variable.

Subscale	Dif. Source	Sum of Squares	df	Mean Square	F	Sig.
JCS	Between Groups	2.401	2	1.200	2.702	0.076
	Within Groups	23.995	54	0.444		
	Total	26.396	56	-		
GWB	Between Groups	2.057	2	1.029	3.031	0.057
	Within Groups	18.324	54	0.339		
	Total	20.381	56	-		
HWI	Between Groups	7.141	2	3.570	4.673	0.013 **
	Within Groups	41.257	54	0.764		
	Total	48.398	56			
CAW	Between Groups	5.784	2	2.892	5.284	0.008 **
	Within Groups	29.553	54	0.547		
	Total	35.337	56	-		
WCS	Between Groups	5.608	2	2.804	6.584	0.003 **
	Within Groups	23.000	54	0.426		
	Total	28.608	56	-		
SAW	Between Groups	4.436	2	2.218	2.756	0.072
	Within Groups	43.458	54	0.805		
	Total	47.895	56	-		
Overall WRQoL-2 Score	Between Groups	2.996	2	1.498	4.922	0.011 **
	Within Groups	16.436	54	0.304		
	Total	19.432	56	-		

** Significant at 0.01.

**Table 7 healthcare-10-02539-t007:** Scheffe’s post ad hoc test for three subscales.

Subscale	Age	n	Mean	Under 25	25 to 44	45 to 59
HWI	Under 25	3	2.78	** - **		
	25 to 44	48	3.19		** - **	
	45 to 59	6	4.28	******		** - **
CAW	Under 25	3	1.89	** - **		
	25 to 44	48	3.17		** - **	
	45 to 59	6	3.56	** ** **		** - **
WCS	Under 25	3	2.56	** - **		
	25 to 44	48	3.58		** - **	
	45 to 59	6	4.22	** ** **		** - **
Overall WRQoL-2 Score	Under 25	3	2.84	** - **		
	25 to 44	48	3.26		** - **	
	45 to 59	6	3.92	** ** **		** - **

** Significant at 0.01.

**Table 8 healthcare-10-02539-t008:** Comparison of WRQoL-2 Scale Across Multiple Studies.

Study	JCS	CAW	HWI	GWB	SAW	WCS
Current Study (MRITs) in KSA	3.59	3.15	3.28	3.51	3.42	3.6
Zhao et al. (2013) (Nurses in China) [24]	3.48	3.39	3.52	3.44	3.62	3.62
Edwards et al. (2019), (Higher Education Staff) [25]	3.43	3.39	3.52	3.44	3.62	3.62
Easton et al. (2013), (Police in the UK) [20]	3.09	2.98	2.77	3.12	2.6	2.81
Mazloumi et al. (2014) (Train drivers in Iran) [26]	3.21	3.04	2	3.62	4.29	1.37
Kahyaoglu Sut& Mestogullari (2016) (Nurses without PMS Turkey) [21]	3.3	3.4	3.1	3.3	2.8	2.9
Kahyaoglu Sut & Mestogullari (2016), (Nurses with PMS Turkey) [21]	3	2.9	2.6	2.7	2.5	2.4
Opollo et al. (2014), (Ugandan Nurses) [23]	3.53	-	2.46	-	-	-
Dai et al. (2016) (Nurses in Taiwan) [22]	3.57	3.39	3.42	3.25	3.46	3.33

## Data Availability

Quantitative data for this study is available by contacting the corresponding author.

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
