# Peer review of "Quality of Work Life of Magnetic Resonance Imaging Technologists: A Cross-Sectional Study"

_healthcare, 2022, doi:10.3390/healthcare10122539_

Round 1
Reviewer 1 Report
Quality of Work Life (QWL) is a multi-dimensional discipline that is concerned with the quality of life (QoL) at the workplace.
The authors aimed to assess the QWL level and identify the correlation between QWL dimensions and Job and Career Satisfaction (JCS).
They used the 32-item WRQoL-2 tool, a questionnaire consisting of 6 subscales: Job and Career Satisfaction (JCS), Control at Work (CAW), Home-Work Interface (HWI), General Well-Being (GWB), Stress at Work (SAW), and Work Conditions (WCS).
57 Magnetic Resonance Imaging Technologists (100%) responded to the questionnaire.
They found a high level of QWL among Magnetic Resonance Imaging Technologists (66.2%, 3.31/5). The level of the JCS was high (71.6%, 3.59/5), with a significant correlation between the JCS and WCS, CAW, HWI, and GWB. An inverse relationship was noted between SAW and JCS.
They concluded that further research on QWL is advised to diagnose and provide recommendation to resolve issues that may adversely affect the quality of healthcare service provision.
Interesting study.
I have some minor suggestions with a pure academic spirit:
1. The acronym MRIT is not resolved in the abstract
2. I would suggest to do not use the acronyms (ten in total) in the abstract to improve the readness
3. Insert the limitation in the discussion
4. Add a table/list with the acronyms
5. Avoid the use of “we” and “our”
Author Response
Thank you for your detailed comments and feedback. Please see point-by-point response attached here.

Reviewer 2 Report
Review of Manuscript # Healthcare-2100407
In this work, the authors report the QWL level and identify the correlation between QWL dimensions and Job and Career Satisfaction (JCS). The authors have studied, by using the 32-item WRQoL-2 tool, a questionnaire consisting of 6 subscales. A high level of QWL among MRITs (66.2%, 3.31/5) was found.
The work is appreciated but there is few comments for the improved quality of the present work. In my opinion, the manuscript Healthcare-2100407 should be considered as as minor revision.
Questions and comments:
1. Abstract – the conclusion presented in the Abstract section is more a perspective. In my opinion, the words “Backgroung”, “Methods”, “Results” and “Conclusion” from the abstract, should be deleted;
2. Page 1, line 24: as the satisfaction degree;
3. Page 4, line 185: are younger than 25 years old;
4. Page 7, line 234: please delete the space after “age.”;
5. Page 8, line 248: please delete the space after “[27].”;
6. Page 9, line 266: differ slightly;
7. Page 9, line 285: please delete the space after “[thus,]”.
Author Response
Thank you for your detailed comments and feedback. Please see attached for point-by-point response.
